# Lessons Learnt from Rapid Implementation of Telehealth in a Paediatric Dietetics' Outpatient Service: Is There a Silver Lining beyond the Coronavirus Pandemic to Support Patient-Centred Care?

Araceli Llanos Jeri [1], Kelly Lambert [1], May Mak [1,2] and Marika Diamantes [2,*]

1 School of Medical, Indigenous and Health Sciences, University of Wollongong, Wollongong 2522, Australia
2 Department of Dietetics, Liverpool Hospital, Liverpool 2170, Australia
* Correspondence: marika.diamantes@health.nsw.gov.au

**Abstract:** The aim of this paper is to report on the differences in clinical and service engagement outcomes of patients attending the paediatric dietetic outpatient service at a major metropolitan hospital before the outbreak of the coronavirus (using face-to-face care) and during the pandemic (using telehealth). This paper also reports on learning lessons from the rapid implementation of telehealth in this service. This study collected pre- and post-coronavirus pandemic data from 44 paediatric patients. Data on outcomes pre- and post-implementation were analysed. There were no statistically significant differences between pre- and post-coronavirus data for growth ($p = 0.92$), adherence to dietetic recommendations ($p = 0.08$) or attendance rate ($p = 1.00$). This study also found a low uptake of virtual telehealth, technical issues and suboptimal anthropometric data collection associated with this modality of care. Telehealth was not associated with a change in clinical and service engagement outcomes. Thus, telehealth service delivery is not inferior to usual face–face dietetic care and has the potential to be a useful adjunct to usual nutrition care for paediatric health service users after the coronavirus pandemic.

**Keywords:** telehealth; coronavirus; paediatric; cross-sectional survey; dietitian; dietetic



## 1. Introduction

The Coronavirus (COVID-19) pandemic is a global health crisis that has affected the way health services are accessed. Social distancing and quarantine measures were used to reduce the transmission of COVID-19 and led to the unprecedented, mass-scale and rapid deployment of telehealth as a service delivery option in the healthcare system [1]. The rapid implementation of telehealth has been reported in various clinical areas across the world [2,3].

Telehealth is the "delivery of health care services through the use of information and communication technologies" (ICT) [4]. The principal modes of telehealth delivery today include telephones (audio), video conferencing (audio-video), mobile applications and short messaging services or SMS [5]. Since the COVID-19 pandemic, many studies have reported on the benefits and challenges of implementing telehealth. The reported benefits of telehealth include scheduling flexibility, reduced transportation costs, improved access to health services, continuity of person-centred care, preserved use of personal protective equipment (PPE) and compliance with social distancing advice (the latter two being imperative during the pandemic) [4,6–9]. The negative aspects of telehealth previously documented include a lack of compliance and interest from patients, inability to conduct a physical assessment, technical issues such as problems with the internet and difficulty in building rapport online [6,9]. In addition to the negative aspects, there has also been a documented lack of uptake from health care providers (HCP), especially allied health [9].

Reasons for this included the HCPs' unwillingness to embrace telehealth, technological problems experienced by patients, and inability to perform face-to-face services such as physical examinations [9].

In the paediatric clinical context, telehealth is being increasingly used to make feasible, acceptable, and effective multi-disciplinary assessments [10]. In children with feeding difficulties, telehealth offers dietitians the unique opportunity to observe the child and guardian in their home environment rather than in a clinical setting, thus affording an assessment that might be more reflective of usual behaviour while offering parents/carers the benefit of accessing routine dietetic services without having to attend the physical clinic in person [11]. In addition, telehealth use in paediatrics provides the possibility for a greater attendance of patients' caretakers and family members [12]. Telehealth also gives patients and/or carers the opportunity to easily access and show the dietitian the medications, supplements and food products their child takes at home. This provides the dietitian with a personalised teaching opportunity where they can use the patient's own foods and measuring kitchen equipment as examples [12]. However, there are ethical, clinical and legal issues with regard to safety, including guardians needing to be present for sessions and consent, the accuracy of anthropometry measurements, access to measurement equipment at home and problems involving managing behaviour and sustaining patient engagement (for example, a child being distracted during the session by seeing their image on the monitor) [13].

To date, few studies have examined the impact of telehealth in the paediatric context, specifically dietetics. Primarily, telehealth studies have focused on providing guidance for paediatricians in managing children's general health needs, not specifically dietetics needs [14]. A systematic scoping review outlined that future research is needed in the allied health field, particularly dietetics, to determine models for the successful implementation of telehealth services [15,16]. The limited studies that have been brought forward in paediatric dietetics showed positive outcomes as a result of implementing telehealth [17,18]. One study showed the implementation of an interactive telehealth intervention was successful in rural settings in the United States of America (USA), including lowering the body mass index (BMI) of children who participated and assisting families in adapting their eating behaviour [17]. Similar findings were observed in another study conducted among preschool children from Romania, Spain and Sweden [18]. However, these studies are mainly focused on weight management in the paediatric population. This highlights a gap in the literature on the effectiveness of dietetics telehealth intervention for children requiring oral and enteral nutrition support, including in Australia.

In summary, telehealth has been proven to be an acceptable modality of care provision for paediatric patients. However, limitations have also been identified. There is also a lack of evidence on the effectiveness of telehealth on the outcome of children requiring oral and enteral nutrition support. Local feedback from caregivers highlighted the logistic difficulty in attending in-person dietitian appointments (for example, insufficient car parks within close proximity to the service). With the COVID-19 pandemic, health organisations had no option but to implement telehealth to continue the essential services with limited planning. All these highlighted the importance of this current study in filling the gap in existing literature, addressing a local need and evaluating the rapid implementation of telehealth during the COVID-19 pandemic. The aim of this paper is to report on the results of a study investigating the difference in the clinical and service engagement outcomes of patients attending the paediatric dietetics outpatient service before the COVID-19 pandemic (using the face-to-face modality of care) and during the pandemic (using telehealth). This paper will also report on the learnings from this rapid implementation of telehealth in this setting.

## 2. Materials and Methods

The setting for this study was a large 877-bed metropolitan hospital in Sydney. Rapid implementation of telehealth was required for the paediatric dietetics outpatient service at the beginning of the COVID-19 pandemic in March 2020 to address the need to continue a

critical clinical service under strict social distancing regulations. Telehealth appointments included either telephone or online video conferencing using the platform 'Pexip'. Due to the physical distancing requirements, only telehealth was offered to patients and their carers for dietetics appointments during the COVID-19 period and no patients or carers declined this modality of care.

The clinical and service engagement data were obtained from a retrospective audit of clinical records. This study compared paediatric patient groups from the pre-COVID-19 period (1 April 2019 to 31 March 2020) and the post-COVID-19 period (1 April 2020 to 31 March 2021). Data for this study were collected from patient groups that attended the paediatric dietetic outpatient services. This study was part of a larger body of research conducted across the local health district (LHD) to evaluate the impact of allied health services in response to the COVID-19 Pandemic.

The population of this study included all existing and new paediatric patients (under the aged of 18 years) referred to or managed by the Dietetics Services of this study. The patients were seen either in dietitian only or joint multidisciplinary team appointments (with the Dietitian and Speech Pathologist plus/minus Occupational Therapy and Physiotherapist). All of these patients resided in the LHD, had initial and/or review appointments and required oral, enteral or a combination of both of these forms of nutrition support. To be included in the study cohort, participants were required to have at least two appointments in the pre- and/or post-COVID-19 periods.

Clinical outcomes and service engagement data were collected using retrospective documentation audits of the electronic medical record (eMR) and patient appointment booking system by the student dietitian. Data were verified and cross-checked with the paediatric dietitian and head of department. For each appointment (data point), clinical outcomes, such as anthropometry data (weight (kg); length/height (cm); associated growth percentile and growth charts used) and source of anthropometric measurement (parent/carer reported, eMR, clinic), were recorded. If growth chart data were not recorded in the dietitian eMR, percentiles were calculated using the 'Baby Infant Growth Chart Calculator' [19,20] and 'Children Growth Chart Calculator' [21,22]. In addition, growth (weight and length/height) progression was evaluated at each appointment by comparing it with the anthropometric data from the previous session. A patient was categorised as either tracking, if they remained on the same percentile (or higher) on the growth chart for their anthropometric measure compared to the previous appointment or faltering if their anthropometric measure was on a lesser percentile on the growth charts compared to the previous appointment. Lastly, each appointment recorded the patient's compliance to dietetic recommendations by identifying this as a yes (complied with all dietetic goals or 100%), or no (complied with no dietetic goals or 0%).

In addition to clinical outcomes, the following service engagement data were collected: pre- or post-COVID-19 appointment, date of session; reason for dietetic input (oral, enteral or both), type of appointment (initial assessment or review), mode used (face-to-face, telephone, telehealth video) and attendance.

Other demographic data collected from paediatric patients included socio-demographic data such as age (months), gender, country of birth, language spoken at home, whether an interpreter was required for parent or carer of the paediatric patient and medical background.

All categorical variables are presented as *n* (%), and all continuous variables are reported as median (IQR). All data were compared between the pre- and post-COVID-19 period using the Pearson Chi Square Test or Fisher's Exact Test for categorical variables and the Non-Parametric Independent Samples T-test for continuous variables. $p < 0.05$ was considered statistically significant. Prior to conducting the statistical tests for all the continuous variables, the Shapiro–Wilk Test was performed where a value of $p < 0.05$ indicated data were not normally distributed, and thus, a non-parametric test was used. All statistical analyses were performed using IBM SPSS Statistics 25.0 (SPSS Statistical Package for the Social Sciences, Stanford, CA, USA).

To ensure patient confidentiality, data collected were kept in de-identified format. Data were stored on the secure network drive in a password-protected folder, accessible only by the investigators. Ethics approval was granted by the institutional/local review board. The conduct of the research complied with the SQUIRE guideline [23].

## 3. Results

There were 29 eligible participants pre-COVID-19 and 44 participants in the COVID-19 telehealth period. Details of demographic information, including sex, age, nationality, language spoken at home and medical conditions, are presented in Table 1. There was no statistically significant difference between the pre- and post-COVID-19 groups.

**Table 1.** Participant characteristics of paediatric dietetics outpatient clients during the pre- and post-COVID-19 period.

| Characteristics | Pre-COVID-19 (1 April 2019 to 31 March 2020) No. of Participants = 29 | COVID-19 Telehealth (1 April 202 to 31 March 2021) No. of Participants = 44 | *p*-Value |
|---|---|---|---|
| Age (months), median (IQR) | 12 (7–36) | 21 (14–60) | 0.15 |
| Gender, *n* (%) | | | |
| Female | 18 (62%) | 23 (52%) | 0.41 |
| Male | 11 (38%) | 21 (48%) | |
| Country of birth, *n* (%) | | | |
| Australia | 28 (97%) | 43 (98%) | 0.76 |
| New Zealand | 1 (3%) | 1 (2%) | |
| Language, *n* (%) | | | |
| English | 23 (80%) | 37 (84%) | |
| Arabic | 2 (7%) | 2 (5%) | 0.98 |
| Vietnamese | 2 (7%) | 3 (7%) | |
| Other | 2 (7%) | 2 (5%) | |
| Interpreter required for parent/carer, *n* (%) | | | 0.76 |
| Yes | 4 (14%) | 5 (11%) | |
| Medical Background, *n* (%) | | | |
| Congenital | 5 (17%) | 5 (11%) | |
| Failure to Thrive | 3 (10%) | 8 (18%) | |
| Gastrointestinal | 1 (3%) | 2 (5%) | |
| Global Developmental Delay | 4 (14%) | 9 (20%) | 0.80 |
| Genetic Disorders | 8 (28%) | 11 (25%) | |
| Infectious Disease | 0 (0%) | 1 (3%) | |
| Prematurity | 8 (28%) | 8 (18%) | |

IQR: interquartile range.

### 3.1. Clinical Level Outcomes

In the pre period, there were 26 (83%) patients with optimal growth (i.e., tracking of growth along the centiles) compared to 36 (82%) in the post period. There were no significant differences between the pre- and post-time periods for growth tracking (*p* = 0.92). In relation to compliance with the recommendation, only one (3%) participant demonstrated full (100%) compliance with dietetic recommendations in the pre period compared to eight (18%) in the post period. Despite this, there was no statistically significant difference between the two time points (*p* = 0.08). Details of clinical-level outcomes are presented in Table 2.

**Table 2.** Clinical-level outcomes of paediatric dietetics outpatient clients during the pre- and post-COVID-19 period.

| Clinical Outcomes | Pre-COVID-19 (1 April 2019 to 31 March 2020) No. of Participants = 29 | COVID-19 Telehealth (1 April 202 to 31 March 2021) No. of Participants = 44 | *p*-Value |
|---|---|---|---|
| Growth tracking, *n* (%) | | | |
| Yes (tracking) | 24 (83%) | 36 (82%) | 0.92 |
| No (faltering) | 5 (17%) | 8 (18%) | |
| Full (100%) adherence to dietetics recommendations, *n* (%) | | | 0.08 |
| Yes (100% compliance) | 1 (3%) | 8 (18%) | |
| No (<100% compliance) | 28 (97%) | 36 (82%) | |

IQR: interquartile range.

### 3.2. Service Level Outcomes

The median attendance rate was 100% (IQR: 93–100) in the pre period and 100% (76.25–100) in the post period. There were no statistically significant differences between pre- and post-COVID-19 data for attendance rates (*p* = 1.00).

### 3.3. Telehealth (Post-COVID-19)

There were 149 telehealth sessions booked for the paediatric dietitian outpatient service during the COVID-19 telehealth period (1 April 2020 to 31 March 2021), which represented 90 percent of the total scheduled dietetic appointments. To further understand the nature of telehealth appointments, characteristics and challenges, such as modality, sources of anthropometric data and reason for non-attendance and for appointment delays/interruptions, are presented in Table 3. Forty percent of the dietitian appointments booked during this time employed the video/virtual component for the telehealth online platform. The remaining telehealth sessions when conducted via the phone or using the audio feature of the online health platform. In relation to the anthropometric measures, most of the weight and length data were reported by the parent/carer and of note, only 33 percent of the appointments documented length/height data. Various technical issues, such as device audio/camera not being switched on, audio and/visual component of the platform not working and computer/devices freezing during a session, were experienced during telehealth consults, and led to either a did not attend (DNA) or delay in appointments. Difficulties appeared to happen on both the patient's and clinician's ends.

**Table 3.** Characteristics and challenges of paediatric dietetics outpatient clients during the post-COVID-19 period (1 April 202 to 31 March 2021).

| Domains | Characteristics |
|---|---|
| Modality | Total number of scheduled appointments (*n* = 149) <br> Total number of attended appointments (*n* = 139) <br> • Audio—phone (49%) <br> • Audio—telehealth platform (2%) <br> • Virtual—telehealth platform–dietitian only (25%) <br> • Virtual—telehealth platform–multidisciplinary clinic (15%) <br> • Face to face (9%) |

**Table 3.** *Cont.*

| Domains | Characteristics |
|---|---|
| Anthropometric measurement | Weight recorded: recorded for *n* = 120 (81%) of appointments<br>Sources of weight data<br><br>• Parent/carer report (54%)<br>• Clinic scale (22%)<br>• Electronic medical record (17%)<br>• Paediatric nurse clinic scale (5%)<br>• Dietitian handover (2%)<br><br>Height recorded: recoded for *n* = 51 (33%) of appointments<br>Source of length data<br><br>• Parent/carer report (39%)<br>• Electronic medical record (29%)<br>• Clinic scale (20%)<br>• Paediatric nurse clinic scale (10%)<br>• Dietitian handover (2%) |
| Reason for failure to attend telehealth appointment | • Parent/carer forgot or was unaware of appointment/inconvenient time *n* = 3<br>• Patient in hospital *n* = 2<br>• Not answering phone calls/answering with insufficient time to complete assessment *n* = 2<br>• Technical difficulties *n* = 1<br>• Parent/carer and patient difficulties connecting with appointment *n* = 1<br>• Nil reason provided *n* = 1 |
| Reason for delay or interruption to telehealth appointment | • Technical issues *n* = 3<br>• Clinician delay *n* = 1<br>• Fire alarm *n* = 1<br>• Parking issues *n* = 1 |

Abbreviation: *n* = number of patients.

## 4. Discussion

The purpose of this study was to assess whether telehealth provides comparable clinical and service engagement outcomes as the traditional face-to-face paediatric dietetic appointment. This study demonstrated that telehealth service delivery to paediatric dietetic clients was not associated with changes in clinical and service level outcomes, including growth ($p = 0.92$), compliance to dietetic recommendations ($p = 0.08$) and attendance rate ($p = 1.0$). Other findings were that telehealth is not without challenges, and these require consideration before widespread rollout.

The findings regarding non-inferiority are supported by other studies. For example, an Australian cross-sectional community survey reported that 61.9% of their participants aged 18 years and over, rated their telehealth experience to be "just as good" as their traditional face-to-face care [24]. Another study also reported 85% of Australian adults found their telehealth appointments to be similar or more effective compared to their face-to-face consults [25].

Regarding compliance, this study showed there was a higher percentage of participants reaching full compliance with dietitian recommendations post COVID-19. This difference did not reach statistical significance. However, it may still have clinical importance. This finding is supported by other studies, which reported that levels of goal attainment, and thus, compliance to dietetic recommendations were the same for children in outpatient dietetic services regardless of the mode of delivery, in-clinic or telehealth [11]. Achieving these levels of compliance could have been because telehealth offers dietitians a platform to better identify barriers at home that could impact the child's ability to meet their dietetic goals [26].

In terms of service level outcomes such as attendance rate, it was clear that telehealth in the post-COVID-19 period delivered very similar results to that of face-to-face consults. It is possible that the advantages of telehealth such as convenience, time efficiency, the comfort of being at home, cost-effectiveness and easy access assisted participants to maintain appointment attendance [27]. Studies have also demonstrated that approximate face-to-face attendance rates in Australian children's hospitals are 95% and 90.3% in Australia's general practices [28,29]. These rates are closely reflected in this research, with 100% median attendance rates for both pre and post periods. Unfortunately, patterns of attendance during telehealth service delivery for children under 18 years are still an under-researched area.

This study also further investigated the nature of telehealth appointments. Despite the term "telehealth" being used, the majority of telehealth appointments were conducted using the phone, and only 40% of consults were via the virtual platform. This indicated that there is still hesitancy from the parents/carers in adopting virtual telehealth.

Obtaining anthropometric data is the main concern and challenge for clinicians when implementing telehealth. This is particularly important for paediatric populations as small changes in weight and length can have a significant impact on clinical management. This study highlighted that length data were only collected for about one-third of the appointments and that the majority of the anthropometric data was self-reported by parents/carers. This brings legitimate concerns to clinicians about the accuracy of self-report data. Other sources, however, were also identified, including the eMR or results from other clinicians, both of which have been described as effective data collection methods [12]. This is supported by a recent study that demonstrated that nutrition assessment data can be acquired with good precision during telehealth consults via the use of eMR, patient interviews and handover from other healthcare professionals [12]. In addition to the challenges in obtaining anthropometric data, this study also highlighted that technical difficulties still remained as one of the barriers to implementing a successful telehealth session.

The rapid implementation of telehealth in the paediatric dietetics outpatient clinic allowed ongoing clinical service delivery during the COVID-19 pandemic. This situation also inadvertently created an opportunity to investigate how telehealth care compares to standard face-to-face service delivery.

To assist with telehealth delivery, this study suggests several recommendations. Firstly, it is important to note that the recent 2020 New South Wales (NSW) public hospitals outpatient survey reported that 31% of younger patients described their telehealth experience as 'not as good as their face-to-face appointments' [27]. Another Australian survey also reported younger Australians were least likely to have used telehealth during the pandemic [25]. To address this, it is recommended that dietitians offer parents or carers of paediatric outpatients the opportunity to choose between in-person and virtual appointments. This will ensure patient-centred and continuity of care for these clients. Next, to reduce the DNA rate, it is recommended that healthcare professionals adopt the short message service (SMS). In fact, a study showed a 5.4% reduction in non-attendance in outpatient services at Melbourne's Royal Children's Hospital following the use of SMS [28]. In a population that has a culturally and linguistically diverse (CALD) community, this strategy could be very effective. This is because studies have demonstrated that SMS reminders result in a reduction in DNA rates for non-English speaking patients and English speakers, 8.33% versus 4.1%, respectively [28]. Regarding virtual nutrition-focused physical examinations, it is recommended that whilst telehealth cannot take the place of face-to-face examinations [12], it may be necessary for dietitians to teach carers or parents of paediatric patients the skill of how to correctly take anthropometry measurements at home as this will produce more reliable growth tracking data for the patient [30,31]. This skill can be taught to parents or carers using pre-visit videos or via teach back during telehealth appointments [30,31]. It is also worth noting that telehealth may not be suitable for patients who are just initiating or transitioning into home enteral nutrition (HEN). This is primarily because patients and carers initially need both emotional support and guidance to succeed with HEN [32]. An initial face-to-face appointment in this instance can provide

the opportunity for the parent/carer to observe the dietitian/nurse deliver a feed as well as see the clarity that the feeding tube should be to ensure their child's safety.

As a result of this work, we have outlined the strategies that facilitate an effective telehealth service from the patient and clinician perspectives (Table 4). In addition, we highlight some specific considerations for health service managers when implementing telehealth as standard practice (Table 5).

**Table 4.** Strategies to facilitate effective telehealth sessions.

| Patient Perspectives | Clinician Perspectives |
|---|---|
| • Test check-in and telehealth connection prior to the appointment to ensure the microphone and video camera are working.<br>• Ensure devices are fully charged (or plugged in)<br>• Be familiar with dial-in instructions prior to the appointment.<br>• Ensure the environment for the call is private and with minimal distractions.<br>• Ensure a troubleshooting contact is available for example another family member.<br>• Be open to a new way of attending appointments.<br>• Ensure anthropometric measurements (including length/height and weight) are available or taken prior to the appointment. | • Allocate longer appointment times, especially for the first consult.<br>• Suggest 75 min for new and 45–60 min for review appointments (the upper range would be suggested when interpreters are needed for the session)<br>• Ensure telehealth information is clear, concise, and user-friendly. It is suggested to write these materials at reading grade level 8 or lower and use images to support instructions.<br>• Ensure adequate administrative support is available, including the following:<br>  ○ Staffing to design and send out clear instructions;<br>  ○ Staffing to send out troubleshooting instructions;<br>  ○ Staffing to send out reminders via phone call/message;<br>  ○ Staffing to provide troubleshooting support.<br>• Provide detailed instructions on taking anthropometric measurements prior to appointments.<br>• Liaise with other health professionals to obtain anthropometric measurements. |

**Table 5.** Specific considerations for Health Service Managers to facilitate successful incorporation of telehealth as standard care.

| Consideration for Health Service Managers |
|---|
| • Identify a "champion" clinician who can provide guidance and resources to others.<br>• Ensure staff have timely online training on the use of the telehealth platform.<br>• Advocate for the development of a telehealth platform that closely resembles the common commercial platforms, for example., Zoom. These may be more familiar to use for patients than health-specific platforms.<br>• Ensure staff have access to Occupational Health and Safety advice and support on appropriate workstation set up to minimise the risk of developing issues such as eye strain and neck, back, and wrist pain, particularly with increased desk and screen time, associated with using telehealth.<br>• Ensure suitable technology is available for staff, including computers, cameras, and noise-cancelling headsets.<br>• Ensure staff have access to a private quiet area to conduct telehealth consultations.<br>• Establish local procedures or guidelines on the effective use of telehealth, including cyber security, and processes to ensure patient privacy and confidentiality are maintained.<br>• Actively liaise with multicultural health and interpreter services to increase engagement and support to CALD populations.<br>• Develop resources to support patients to implement an effective telehealth session (including instructions in other languages)<br>• Promote the benefit of telehealth to patients/carers as part of the mixed model of care, for example., reducing the time of travelling and reducing the cost of transport/parking.<br>• Develop clear guidelines on a hybrid model of care, including clear guidelines on when face-to-face appointments are required.<br>• Train staff on the most effective way to utilise interpreter services during telehealth consultations. |

Abbreviation: CALD—culturally and linguistically diverse background.

Beyond the COVID-19 pandemic, it is important to investigate whether there is still ongoing support and willingness from patients and carers in using telehealth. A recent survey of 2080 adults in America found 66.5% preferred at least some form of telehealth in the future. However, when it was a decision between in-person and telehealth, 53% preferred an in-person consult [33]. It is evident that face-to-face appointments are still

preferable. As a part of future planning, it is crucial to identify whether paediatric carers still accept and support telehealth as a routine care post pandemic. Future research should focus on identifying carers' perception of telehealth as a modality of care on its own, irrespective of physical isolation or pandemic. A more in-depth qualitative interview can be conducted to explore carers' experiences of telehealth, including their perceived benefits and challenges. It is also important to seek carers' suggestions and opinions in addressing some of the challenges of telehealth. This insight will be valuable to help HCPs continue to utilise telehealth as a complement to in-person appointments while maintaining a patient-centred approach.

This study has both limitations and strengths. The sample population included members of the CALD community, who are an identified at-risk group within telehealth service delivery due to their limited financial capability and lack of access to healthcare [34]. This meant this study did not exclude the 25–30% of the population that make up the CALD paediatric community. However, it is worth noting that despite having a high population born overseas, only 14% (pre-COVID-19) and 11% (post-COVID-19) of the participant's parents or carers required an interpreter. The sample population also included patients fed enterally and those from a variety of medical backgrounds, both of which are representative of the general outpatient paediatric population with feeding difficulties. This study thus has high external validity, that is, it may be generalisable to all paediatric health service users. Another strength is the inclusion of duplicate data extraction to ensure accuracy. Lastly, it is important to note that the rapid implementation of the telehealth service required experienced clinicians who could readily adopt and adapt to this new model of care.

There were also limitations to this study. Firstly, limitations may have arisen in terms of self-selection bias. That is, participants who were more comfortable with telehealth may have been more likely to accept telehealth appointments in the post-COVID-19 period. The sample thus may consist of a post-COVID-19 population who have potentially higher levels of digital literacy. Second, due to the retrospective nature of this study, there may also have been data inconsistencies due to unclear documentation in the patient eMR and electronic booking system. Third, this study did not investigate parents' or dietitians' perspectives regarding telehealth service delivery. Further studies thus should investigate parents'/carers' and dietitians' attitudes towards and experiences with telehealth, as they are both crucial stakeholders in discussions about the future of telehealth in Australia [24]. These studies can significantly benefit the future provision of telehealth services by informing telehealth research and policy within the paediatric field. Future studies should utilise the theoretical domain framework to explore implementation challenges more rigorously. This will broaden the limited evidence base on telehealth use in dietetics [7] and may lead to improved patient-centred care by improving accessibility and inclusiveness.

## 5. Conclusions

Overall, this study found that telehealth service delivery in the paediatric dietetics context was comparable to that of face-to-face care. The results indicated that telehealth was not associated with inferior clinical and patient service level outcomes, including growth, compliance to dietetic recommendations and attendance rate. However, there were concerns with the reliability of the anthropometric measurements, low uptake of virtual telehealth and technical difficulties associated with this mode of care. Despite these challenges, telehealth can be a valuable mode of healthcare delivery and thus a useful addition to traditional care after the COVID-19 Pandemic.

**Author Contributions:** A.L.J.: acquisition, analysis, and interpretation of data for the work and drafting the publication; K.L.: supervision on the analysis, interpretation of data and contribution to the writing of publication; M.M.: Substantial contributions to the conception or design of the work, contribute to the writing of the publication; M.D.: Substantial contributions to the conception or design of the work, supervision of the acquisition and analysis of data, and contribute to the writing of the publication. All authors have read and agreed to the published version of the manuscript.

**Funding:** This research received no external funding.

**Institutional Review Board Statement:** This study was conducted in accordance with the Declaration of Helsinki and has been approved by South Western Sydney Local Health District Human Research Ethics Committee (2020/ETH01959).

**Informed Consent Statement:** This study was approved by the South Western Sydney Local Health District Human Research Ethics Committee (2020/ETH01959). Given that the project methodology involved a retrospective file audit of clinical data from routine services, a waiver of consent was approved in keeping with the Australian National Statement on Ethical Conduct in Human Research Section 2.3.9 and 2.3.10 (https://www.nhmrc.gov.au/sites/default/files/documents/attachments/publications/National-Statement-Ethical-Conduct-Human-Research-2023.pdf).

**Data Availability Statement:** The data presented in this study are available on request from the corresponding author. The data are not publicly available due to privacy reasons.

**Acknowledgments:** Elise Baker assisted in defining the study design, statistical tests, and variables for data collection.

**Conflicts of Interest:** The authors declare no conflict of interest.

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
