# Peer review of "Lessons Learnt from Rapid Implementation of Telehealth in a Paediatric Dietetics’ Outpatient Service: Is There a Silver Lining beyond the Coronavirus Pandemic to Support Patient-Centred Care?"

_2674-0311, doi:10.3390/dietetics2030019_

Round 1

Reviewer 1 Report

Thank you for the opportunity to read and review this paper. This is an interesting study. I suggest the following:

·        Justification is poor. Authors should explain more thoroughly why this study is needed and why it is timely. Authors wrote that there are a few studies on the subject but have not presented the findings from these studies and have not explained what gaps their study is going to fill in.  

·        It is not clear how participants were selected, and if selected randomly. If not, why?

·       Results are presented well.

·        It is good that recommendations are presented in detail and that study’s limitations are discussed.

Reviewer 2 Report

Thank you for this work on an increasingly popular way to connect with patients. With children, this possibility of contact is limited by the interest/ability of the parents.

Here are my comments:
I would recommend in the methods and materials section to better specify what the pre- and post covid phase is. I would also transfer this to the tables.

Do the authors have data from children who did not participate in the Telehealth-program? Would it be possible to use these data as a comparison group?

What is meant by the other recommendations in Table 2?

Table 3 Would it be possible to explain the data by the characteristics of the study population?

Reviewer 3 Report

This study compared telehealth service delivery data for paediatric dietetics during the coronavirus pandemic with pre-pandmeic data and found that the service was equal to face-to-face care, with potential for post-pandemic use. The researchers took advantage of the natural experiment created by pandemic restraints, and the results indicate the importance of continuing the use and development of telehealth services post pandemic.

On Line 241, the authors write:
Beyond the COVID-19 pandemic, it is important to harness the positive aspects of telehealth and continue to address the challenges of remote assessment.  It is important for health providers to continue utilising telehealth to compliment the face-to-face appointments.
My comment is:
This issue requires more in-depth study and discussion in the paper.  Now that the pandemic has passed, will the public continue to show the same degree of support for telehealth?  Are you assuming that public attitudes will remain the same moving forward?  How will the public's tolerance and acceptance of alternative services provided under emergency conditions change once the emergency has ceased?  Please discuss this issue in more detail and provide recommendations for further study in this area.

Other minor comments in the attached pdf.
